# The Role of Salivary Vascular Endothelial Growth Factor A, Cytokines, and Amino Acids in Immunomodulation and Angiogenesis in Breast Cancer

**DOI:** 10.3390/biomedicines12061329

**Published:** 2024-06-14

**Authors:** Elena A. Sarf, Elena I. Dyachenko, Lyudmila V. Bel’skaya

**Affiliations:** Biochemistry Research Laboratory, Omsk State Pedagogical University, 14, Tukhachevsky Str., 644099 Omsk, Russia; sarf_ea@omgpu.ru (E.A.S.); dyachenko.ea@gkpc.buzoo.ru (E.I.D.)

**Keywords:** saliva, VEGF, pro-inflammatory cytokine, amino acids, breast cancer, angiogenesis, immunomodulation

## Abstract

In this work, we focused on the analysis of VEGF content in saliva and its relationship with pro-inflammatory cytokines and amino acids involved in immunomodulation and angiogenesis in breast cancer. The study included 230 breast cancer patients, 92 patients with benign breast disease, and 59 healthy controls. Before treatment, saliva samples were obtained from all participants, and the content of VEGF and cytokines in saliva was determined by an enzyme-linked immunosorbent assay, as well as the content of amino acids by high-performance liquid chromatography. It was found that VEGF was positively correlated with the level of pro-inflammatory cytokines IL-1β (*r* = 0.6367), IL-6 (*r* = 0.3813), IL-8 (*r* = 0.4370), and IL-18 (*r* = 0.4184). Weak correlations were shown for MCP-1 (*r* = 0.2663) and TNF-α (*r* = 0.2817). For the first time, we demonstrated changes in the concentration of VEGF and related cytokines in saliva in different molecular biological subtypes of breast cancer depending on the stage of the disease, differentiation, proliferation, and metastasis to the lymph nodes. A correlation was established between the expression of VEGF and the content of aspartic acid (*r* = −0.3050), citrulline (*r* = −0.2914), and tryptophan (*r* = 0.3382) in saliva. It has been suggested that aspartic acid and citrulline influence the expression of VEGF via the synthesis of the signaling molecule NO, and then tryptophan ensures tolerance of the immune system to tumor cells.

## 1. Introduction

Normal angiogenesis is maintained in homeostasis by numerous pro- and anti-angiogenic factors, resulting in a normal rate of blood vessel growth [1]. During tumor angiogenesis, both tumor cells and tumor-associated stromal/immune cells secrete proangiogenic factors [1]. Vascular endothelial growth factor A (VEGF) is the most potent proangiogenic factor [2]. VEGF is known to be secreted by inflammatory cells (activated neutrophils, monocytes/macrophages, T cells), as well as dendritic cells, platelets, endothelial and tumor cells [3,4]. VEGF mediates angiogenesis, vascular permeability, and inflammation by binding to the high-affinity tyrosine kinase receptors VEGFR1 and VEGFR2, expressed mainly on vascular endothelium but also on tumor cells [5,6].

The complex cell matrix of breast tumors includes stromal cells and adipocytes. It is known that up to 50% of breast tumors may consist of immune system cells infiltrating the tumors. These include tumor-associated macrophages (TAMs) and tumor-infiltrating lymphocytes (TILs), which are attracted by chemokines such as IL-8 and macrophage chemoattractant protein-1 (MCP-1) secreted by tumor cells. An abundant blood supply is essential for tumor growth, and TAMs may be an important source of angiogenic factors that stimulate the development of blood vessels in breast tumors [7]. Cytokines, particularly IL-6 and tumor necrosis factor-α (TNF-α), are also known to play a critical role in regulating estrogen synthesis in breast cancer cells [8].

VEGF, cytokines, and their relationship with breast cancer have been well studied, but tumor tissue or blood are mainly used as biomaterials [9,10,11,12,13,14,15]. The content of VEGF in the tumor and blood plasma correlates with the prognosis of breast cancer, while the concentration in saliva has been practically not studied to date. Previously, we examined the content of pro-inflammatory and anti-inflammatory cytokines in saliva in breast cancer [16]. Salivary cytokine levels have been shown to correlate well with clinicopathological and molecular biological characteristics of breast cancer. However, we previously considered a group of patients with benign breast pathologies as a comparison group, whereas in this work, we added healthy controls.

It is known that amino acids play an important role in the progression of breast cancer as sources of building material for cancer cells, participants in the immunomodulation of the tumor microenvironment, and activators of angiogenesis [17]. Thus, the level of VEGF expression is regulated by the concentration of nitric oxide (NO), which in turn directly depends on the concentration of L-arginine and additional amino acids citrulline (Cit) and aspartic acid (Asp) [18]. It is known that tryptophan (Trp) promotes active cell proliferation by activating cell cycle signaling pathways and also inhibits T-cell differentiation, which induces tolerance of the immune system to cancer cells [19,20]. Amino acids such as glutamine (Gln), glycine (Gly), and proline (Pro) perform a wide range of functions, from the synthesis of nucleotides, proteins, and glutathione to DNA methylation and histones, and may be involved in the immune response in breast cancer [21,22,23,24,25,26].

In this work, we focused on the analysis of VEGF content in saliva and its relationship with pro-inflammatory cytokines and amino acids involved in immunomodulation and angiogenesis in breast cancer.

## 2. Materials and Methods

### 2.1. Study Design

The study included 381 volunteers, including 230 patients with breast cancer (main group, age 60.0 [47.8; 66.8] years), 92 patients with non-malignant pathologies of the mammary glands (comparison group, age 44.7 [38.8; 57.0] years), and 59 healthy controls (control group, age 44.9 [36.1; 52.7] years). Patients in the main and comparison groups were hospitalized for surgical treatment, and after histological verification, they were assigned to one of the groups. The control group included volunteers who had no mammary gland pathologies detected during mammography. Determination of VEGF and cytokines was carried out in all study participants (n = 381), while the determination of amino acids was carried out in 116 patients of the main group, 24 patients with non-malignant pathologies of the mammary glands, and 25 healthy controls. Table 1 shows a detailed description of the subgroups of breast cancer patients for whom VEGF, cytokine, and amino acid analyses were performed.

Inclusion criteria: female gender, patient age 30–60 years, absence of any treatment at the time of the study, including surgery, chemotherapy, or radiation, absence of signs of active infection (including purulent processes). All participants were examined by a dentist and had good oral hygiene. Exclusion criteria: lack of histological verification of the diagnosis.

### 2.2. Collection of Saliva Samples

Saliva samples were collected during hospitalization strictly before the start of treatment. Samples were collected in sterile polypropylene centrifuge tubes with a screw cap in a volume of 2 mL. Saliva samples were collected by spitting without additional stimulation in the interval of 8–10 a.m., the time of maximum saliva secretion, on an empty stomach after preliminary rinsing the mouth with water. 

Immediately after collection, samples were centrifuged at 10,000× *g* for 10 min (CLb-16, Moscow, Russia); 1 mL of the upper layer was taken, transferred to Eppendorf tubes, and stored in a freezer at −80 °C until analysis.

### 2.3. Determination of the Amino Acid Composition of Saliva

The content of VEGF (mU/mL) and cytokines in saliva (IL-1β, IL-6, IL-8, IL-18, TNF-α, and MCP-1; pg/mL) was determined by an enzyme-linked immunosorbent assay using Vector Best kits (Russia). In all cases, the same volumes of saliva aliquots (100 μL) were used. We used calibration and control samples based on human blood serum, certified against cytokine and chemokine standards (R&D Systems, Inc., Minneapolis, USA), according to the instructions given in the corresponding reagent kits without changes, including reagent volumes and incubation time. Reading was performed using Thermo Scientific Multiskan FC (Waltham, MA, USA).

### 2.4. Determination of the Amino Acid Composition of Saliva

Concentrations of 26 amino acids were quantified in all saliva samples. Amino acid analysis of the samples was carried out using high-performance liquid chromatography on a 1260 Infinity II chromatograph (Agilent, Santa Clara, CA, USA) with detection on a 6460 Triple Quad mass spectrometer (Agilent; USA). The samples were separated by liquid chromatography using an Agilent Zorbax Eclipse XDB-C18 2.1 × 100 mm column with a sorbent diameter of 1.8 μm (Agilent; USA). To analyze the test compounds in the samples, an HPLC method with mass spectrometric detection in the monitoring mode of selected reactions has been developed. The internal standard method (alanine-d4) was used to back-calculate concentrations. To construct a calibration scale, at least 6 samples of the Amino Acid kit (Jasem, Turkey) were used. To analyze the results, we used automatic integration of chromatograms using the Quantitative Quant-my-way software (MassHunter Workstation Quantitative Analysis B.09.00) (Agilent, Santa Clara, CA, USA).

### 2.5. Determination of the Expression of the Receptors for Estrogen, Progesterone, HER2 and Ki-67

The Allred Scoring Guideline was used to assess the level of expression of estrogen receptors (ER), progesterone receptors (PR), and HER2 [27]. The level of expression of estrogen, progesterone, and HER2 receptors was assigned to one of four categories (−, +, ++, and +++) in accordance with the ASCO/CAP recommendations [28]. Ki-67 expression was determined as part of a standard breast cancer panel according to the manufacturer’s protocol [29]. The cut-off value for Ki-67 was defined as 20% (<20%—low Ki-67; >20%—high Ki-67). According to the obtained values, breast cancer was classified into five groups: triple-negative breast cancer (TNBC), luminal A-like, luminal B-like (HER2-negative), luminal B-like (HER2-positive), and HER2-enriched (Non-Luminal).

### 2.6. Statistical Analysis

Statistical analysis was performed using Statistica 13.3 EN (StatSoft, Tulsa, OK, USA) programs using a nonparametric method. When comparing two groups, we used the Mann–Whitney test; when comparing three groups or more, we used the Kruskal–Wallis test. The sample was described using the median (Me) and interquartile range in the form of the 25th and 75th percentiles [LQ; UQ]. Differences were considered statistically significant at *p* < 0.05.

To establish relationships between VEGF, cytokines and amino acids, the Spearman correlation coefficient was calculated. Correlations were considered strong when the correlation coefficient value was more than 0.7; when the coefficient value was in the range of 0.3–0.7, correlations were considered to be of medium strength; correlations less than 0.3 were considered weak.

## 3. Results

### 3.1. Salivary VEGF Content in Breast Cancer

During the first stage of the study, we compared the content of VEGF in saliva depending on the presence/absence of mammary gland pathologies in comparison with healthy controls. It has been shown that the concentration of VEGF statistically significantly increased in breast cancer (1297.4 [586.7; 2119.5] mU/mL), compared with benign breast pathologies (1081.1 [505.4; 1667.4] mU/mL), and compared with the control group (496.1 [352.7; 1360.2] mU/mL) (Figure 1A). Additionally, a comparison was made of VEGF content depending on the clinicopathological and molecular biological characteristics of breast cancer. The greatest increase in VEGF concentrations was observed for early breast cancer (stage I—1397.4 [809.0; 2180.1] mU/mL), followed by a decrease in advanced stages (stage II—1242.2 [445.4; 2123.7]; stage III + IV—1190.7 [794.3; 1943.4] mU/mL) (Figure 1B). The differences with the control group were statistically significant in all cases. It was shown that in the absence of metastases in the lymph nodes, the concentration of VEGF in saliva was higher (1334.1 [533.3; 2158.7] vs. 1270.1 [727.4; 1930.1] mU/mL), while the degree of tumor differentiation did not affect the content of VEGF in saliva. The concentration of VEGF in saliva increased with positive expression of estrogen (1344.7 [735.7; 2130.8] vs. 1145.3 [467.4; 1986.7] mE /mL) (Figure 1C) and progesterone receptors (1375.4 [844.7; 2150.7] vs. 1072.7 [438.4; 1957.1] mU/mL) (Figure 1D), HER2-positive status (1338.7 [727.4; 1986.7] vs. 1270.1 [579.4; 2119.5] mU/mL) (Figure 1E), as well as low expression of Ki-67 (1317.4 [769.4; 2135.3] vs. 1170.6 [445.8; 1986.7] mU/mL) (Figure 1F). It is natural that the maximum concentration of VEGF in saliva was observed in the luminal A molecular biological subtype of breast cancer, while the minimum was observed in TNBC (Figure 1G).

### 3.2. Changes in the Concentration of Pro-Inflammatory Cytokines in Saliva in Breast Cancer

During the next stage of the study, we calculated the correlations between the concentrations of VEGF and cytokines in saliva in breast cancer. We were interested in whether there would be a relationship between the presence of high levels of VEGF and the activation of the immune response in breast cancer. VEGF was shown to positively correlate with the level of pro-inflammatory cytokines IL-1β (r = 0.6367), IL-6 (r = 0.3813), IL-8 (r = 0.4370), and IL-18 (r = 0.4184). Weak correlations were shown for MCP-1 (r = 0.2663) and TNF-α (r = 0.2817).

Next, we analyzed changes in the concentration of pro-inflammatory cytokines, for which correlations with VEGF were identified, in the same subgroups as previously for VEGF (Figure 1).

The mean salivary cytokine levels in breast cancer, non-malignant breast pathologies, and healthy controls are shown in Table 2.

Figure 2 shows the change in salivary cytokine levels compared to healthy controls. It can be seen that the concentration of IL-1β increased in breast cancer and benign breast pathologies, as well as the concentration of VEGF (Figure 2A). For TNF-α, MCP-1, and IL-18, an increase in concentration was shown in breast cancer, while in benign breast pathologies, the concentration remained virtually unchanged compared to healthy controls (Figure 2B). For IL-6 and IL-8, a decrease in concentration was observed compared to healthy controls in both cases (Figure 2B).

For these cytokines, changes in concentration depending on the clinicopathological and molecular biological characteristics of breast cancer were also considered (Figure 3, Figure 4, Figure 5, Figure 6 and Figure 7).

It can be seen that the maximum deviations in cytokine concentrations were observed at stage I of breast cancer: an increase for VEGF, IL-1β, TNF-α, and MCP-1 and a decrease for IL-6 (Figure 3A,B). For IL-8, the opposite trend was observed, and the lowest concentration compared to healthy controls was observed for advanced stages of breast cancer (Figure 3B). For IL-18, there was first an increase in concentration and then a decrease, but these changes were not statistically significant (Figure 3B).

We analyzed the significance of the lymph node lesion status and showed that in the absence of metastases in the lymph nodes, the level of VEGF, IL-1β, and IL-18 was higher, while the level of TNF-α, IL-6, and IL-8 on the contrary, decreased (Figure 4A,B). Only for IL-8 the differences between N_0_ and N_1–3_ were statistically significant (Figure 4B). The concentration of MCP-1 was not affected by the status of lymph node damage (Figure 4B). 

It was shown that highly (GI) and moderately differentiated (GII) tumors had higher levels of MCP-1 and IL-18 in saliva and lower concentrations of IL-8 (Figure 5B). For other indicators, no differences were found between highly and moderately differentiated (GI + II) and poorly differentiated (GIII) breast cancer (Figure 5A,B). For IL-18, opposite changes in concentrations were shown depending on the degree of differentiation; these changes were statistically significant (Figure 5B).

It was shown that, depending on the expression status of HER2, estrogen and progesterone receptors, as well as the index of proliferative activity of breast cancer, the concentrations of IL-1β (Figure 6A), MCP-1 (Figure 6B), TNF-α (Figure 6C), IL-6 (Figure 6D), and IL-8 (Figure 6E) changed unidirectional, while for IL-18, there was a multidirectional change in concentration compared to healthy controls (Figure 6F). Statistically significant differences between positive and negative HER2 receptor status were shown for IL-6. With different expressions of estrogen receptors, the content of IL-18 differed; for progesterone receptors, the levels of IL-1β, IL-8, and IL-18 differed. Differences in the proliferative activity index were reflected in the concentrations of IL-1β and IL-18.

Differences in cytokine concentrations were pronounced between molecular biological subtypes of breast cancer (Figure 7A,B). For the luminal A subtype, the concentrations of VEGF, IL-1β, MCP-1, and IL-18 changed the most. For luminal B (HER2-), the content of IL-6 decreased maximally, while for non-luminal and TNBC, the content of IL-6, IL-8, and IL-18 decreased. The content of IL-18 increased for luminal subtypes of breast cancer and decreased for non-luminal subtypes (Figure 7B).

### 3.3. Relationship between the Content of VEGF, Pro-Inflammatory Cytokines, and Amino Acids in Saliva in Breast Cancer

Previously, we determined the content of 26 amino acids [30]. In this work, correlation coefficients with amino acids involved in major metabolic processes were calculated. It was shown that VEGF correlated with the content of aspartic acid (r = −0.3050), citrulline (*r* = −0.2914), and tryptophan (*r* = 0.3382) in saliva. TNF-α correlated with proline content (*r* = −0.3454), whereas IL-18 correlated with salivary trans-4-hydroxyproline content (*r* = −0.5491). MCP-1 correlated with the content of glutamine (*r* = −0.3266), glycine (*r* = −0.2801), and trans-4-hydroxyproline in saliva (*r* = −0.3310). 

Salivary concentrations of aspartic acid, glycine, and proline were increased compared to healthy controls, whereas glutamine concentrations were decreased (Figure 8A). No significant changes were detected for changes in the concentrations of citrulline, trans-4-hydroxyproline, and tryptophan (Figure 8B).

It should be noted that the change in proline and glutamine concentrations compared to healthy controls was statistically significant in all cases, regardless of stratification by clinicopathological or molecular biological characteristics of breast cancer. At the same time, the concentration of glutamine changed in different molecular biological subtypes ambiguously: it increased in luminal B HER2(+) and TNBC and decreased in all others (Figure 9A). For aspartic acid, differences were shown between HER2(−) vs. HER2(+) subgroups (*p* = 0.0326), which naturally determined the differences between luminal B HER2(+) and luminal A (*p* = 0.0103), luminal B HER2(−) (*p* = 0.0044) and non-luminal (*p* = 0.0102) subtypes breast cancer. For glycine, differences were only shown between HER 2(+) breast cancer and healthy controls (*p* = 0.0261) and, respectively, between luminal B HER2(+) and luminal A (*p* = 0.0472), as well as luminal B HER2(−) (*p* = 0.0114) breast cancer subtypes. For citrulline, the only difference in content was shown between luminal B HER2(+) and non-luminal (*p* = 0.0318) subtypes of breast cancer, with the former being increased compared to healthy controls, while the latter was decreased. Thus, luminal B HER2(+) and TNBC are two molecular biological subtypes of breast cancer in which the concentrations of all amino acids increased compared to healthy controls (Figure 9).

The concentration of Pro in saliva in breast cancer patients was statistically significantly higher than in normal conditions (*p* = 0.0458), while for t4HYP, no differences were found with healthy controls. However, for t4HYP, significant differences were shown when analyzing individual subgroups of breast cancer. Thus, the concentration of t4HYP increased with the increasing stage (St I + II vs. St III + IV, *p* = 0.0417) and degree of differentiation (G I + II vs. G III, *p* = 0.0071) of breast cancer. An increase in t4HYP concentration was observed in the presence of HER2 expression, the negative expression status of estrogen receptors (*p* = 0.0461), and progesterone receptors, as well as high Ki-67.

## 4. Discussion

During the course of our study, we showed an increase in the concentration of VEGF in saliva in breast cancer and also found that changes in the concentration of pro-inflammatory cytokines (IL-1β, IL-6, IL-8, and IL-18) and amino acids (Asp, Cit, and Trp) significantly correlate with changes in the level of VEGF.

We have shown that the content of VEGF in saliva in all considered subgroups had the same pattern as the content of IL-1β. At the initial stages of tumor development, VEGF and IL-1β are important components, as they participate in the formation of a microenvironment favorable to cancer cells by activating angiogenesis, increasing the permeability of the vascular wall, regulating metabolism, modulating the immune system and recruiting pro-inflammatory cytokines [31,32]. We observed an increase in the concentration of VEGF in saliva almost threefold (1297.4 [586.7; 2119.5] vs. 496.1 [352.7; 1360.2] mU/mL), while for serum no more than 1.5 times [33]. These results contradict literature data, according to which the content of VEGF in saliva in normal conditions and in breast cancer does not differ [34]. We also found high levels of VEGF and IL-1β in the early stages of the disease and a high degree of cell differentiation, followed by a slight decrease in their concentrations as the disease progresses. This is explained by the fact that it is the initial period of the disease that is critical in determining the viability of cancer cells due to the presence of VEGF and IL-1β, which determine their further growth and development [35]. When examining the contribution of VEGF to metastasis, it is worth mentioning that VEGF comprises a large family of growth factors that are structurally and functionally similar. This family includes VEGF A, VEGF B, VEGF C, VEGF D, and placental growth factor (PlGF). Thus, VEGF A stimulates angiogenesis and determines the permeability of blood vessels [3,36]. It was shown that the vessels in the tumor microenvironment were shorter, loosely distributed, highly permeable, and lacked septa between capillaries [37,38]. Other isoforms (VEGF C and VEGF D) have an inducing effect on lymph angiogenesis, which leads to metastasis to the lymph nodes [39]. It is important to note that VEGF not only affects the tumor microenvironment locally but also manifests itself at the systemic level [40]. We did not identify a relationship between the VEGF level and metastasis since we measured only the content of salivary VEGF A.

According to the literature, HER2 overexpression is associated with VEGF activation and plays an important role in breast cancer development and metastasis [41,42]. In breast cancer, there is a crosstalk between EGFR and angiogenesis signaling [43], so VEGF is overexpressed in both HER2-negative and HER2-positive breast cancer [44]. Moreover, higher levels of VEGF expression in the tumor were associated with the aggressiveness of breast cancer [45]. Thus, the average level of intratumoral VEGF expression in the TNBC population was significantly higher compared to the non-TNBC population [46]. In our work, we showed, on the contrary, the lowest level of VEGF in saliva in TNBC.

We have shown that the content of IL-1β in saliva in all considered subgroups has the same pattern as the content of VEGF, while for IL-6 and IL-8, an opposite downward trend is observed. Only IL-1β showed statistically significant differences compared to healthy controls. We also observed differences in the content of IL-1β, IL-6, and IL-8 for different molecular biological characteristics of breast cancer. According to the literature, IL-1β promotes the progression of breast cancer by enhancing the migration ability of cancer cells, migratory potential, and disease relapse [47,48]. In addition, it takes part in the activation of HIF-1α, after which the active form of HIF-1α binds to the dimeric transcription factor, which leads to the expression of VEGF [49,50]. Taking into account the anatomical distance of breast cancer cells from the salivary glands and their direct contact only with the blood and lymphatic vessels; saliva, in this case, reflects changes in the concentration of VEGF and IL-1β that occur at the systemic level.

Activation of IL-6 occurs both with increased hypoxia in cancer cells and their microenvironment, and in its absence. Even under non-hypoxic conditions, IL-6 family cytokines have been shown to promote VEGF expression via activation of the transcription factors HIF1-α and STAT3 [51]. IL-6 concentrations are also influenced by IL-1β, acting through transglutaminase 2 (TG2) [52]. In turn, TG2 activates nuclear factor kappa B (NF-kB) and acts as a regulator of the positive feedback loop between NF-kB and IL-6/STAT3, which mediates cancer aggressiveness and hormone-independent tumor growth [53,54]. Commonly, co-expression of VEGF and IL-6 is observed in the blood serum, which serves as markers of negative prognosis, especially in relation to the HER2-negative group [1]. We found a decrease in the concentration of IL-6 among patients with breast cancer compared to the control group. At the same time, the maximum decrease in IL-6 content is observed in luminal B and HER2-negative breast cancer (Figure 7). A number of studies have shown that breast cancer cell lines produce and secrete cytokines of the IL-6 family [55]. In this case, ER+ cells secrete lower levels of IL-6 than ER- cells since estrogen inhibits the expression of IL-6, impairing the transactivation of NF-κB [56,57,58,59]. IL-6 performs pro-inflammatory functions and is associated with aggressive molecular subtypes of breast cancer, as it is an inducer of mesenchymal-epithelial transition [60,61,62], promotes the formation of hormone-independent cancer cells, invasive tumor growth, and metastasis [63]. Plasma levels of IL-6 and IL-8 are higher in breast cancer patients compared to normal healthy donors and are positively correlated with stage and mortality [64,65], which is associated with their potential role in the development of chemoresistance [66].

Our data are difficult to compare with the literature since cytokine determination was previously carried out either in tumor tissue or in blood plasma. Thus, it was shown that the profile of cytokines in the serum in patients with HER2+ was similar to that in patients from the control group, while in patients with HER2-negative status, there was a significantly lower number of cytokines compared to healthy women (IL-6 and IL-8) and cases of HER2+ (IL-1β, IL-6, and IL-8) [67]. However, consistent with our data, we observe a decrease in the content of only IL-6 in HER2-negative breast cancer (Figure 6). Collectively, systemic and local IL-6 expression is known to represent two distinct compartments, with systemic levels reflecting the metabolic and inflammatory status of the entire body, while local IL-6 expression may have a direct effect on cancer cell growth, metastasis and cancer stem cell renewal [68].

Stimulation of VEGF expression occurs not only in the hypoxic state of the cell but also during glucose deprivation [69]. Glucose deficiency triggers endoplasmic reticulum (ER) stress, which leads to the initiation of the unfolded protein response (UPR) signaling pathway and activation of NF-κB [70]. It has been found that in addition to indirectly influencing VEGF activity via glucose deprivation, NF-κB is also an inducer of IL-8 expression in solid tumors [71,72]. This leads to the conclusion that, in addition to the hypoxic state, the lack of nutrients for tumor cells may also serve as a trigger for the proangiogenic tumor phenotype [73]. Increased protein synthesis and IL-8 secretion may also be associated with the estrogen receptor [74], namely ER inactivity, increased cell invasion, and angiogenesis may represent a novel pathway involved in the metastatic progression of HER2-positive breast cancer [75]. IL-8 overexpression in estrogen receptor-negative tumors correlated with breast cancer progression. It is involved in proangiogenic and oncogenic effects on adipocytes, so the IL-8 pathway may have therapeutic value [76]. It was in hormone-negative breast cancers that we also showed the maximum decrease in IL-8 content compared to healthy controls (Figure 7). Our study showed a significant decrease in the concentration of IL-8 in advanced stages of breast cancer (Figure 3), which is apparently due to the presence of regional metastasis (Figure 4), as well as a maximum decrease in IL-8 estrogen- and progesterone-negative status breast cancer and low tumor proliferation (Figure 6). A decrease in salivary IL-8 concentrations was previously shown in a study by Streckfus et al. [34], which is in good agreement with our data. Our study showed a consistent marked decrease in the concentrations of IL-6 and IL-8, which is explained by the uneven redistribution of cytokines toward increased local concentrations in the microenvironment and decreased concentrations at the systemic level.

Particular attention is paid to IL-18, which belongs to the IL-1 family and is an agonist ligand. IL-18 is associated with IL-1β and VEGF, as it can indirectly or directly influence proangiogenic factors [77]. In our study, IL-18 was the only cytokine that was statistically significantly associated with the degree of tumor differentiation (Figure 5). The concentration of IL-18 shows the maximum differences between molecular biological characteristics: it increases with positive expression of estrogen and progesterone receptors and low proliferative activity of breast cancer (Figure 6). Thus, it is natural to note the multidirectional nature of changes in the concentration of IL-18 in hormone-positive and hormone-negative tumors. In the first case, there is an increase in the concentration of IL-18 relative to healthy controls, while in the second case, the concentration decreases maximally in TNBC (Figure 6). Inoue et al. showed that serum IL-18 levels are a significant and independent prognostic factor for disease-free survival among other prognostic factors (including lymph node metastasis, tumor grade, and Ki-67 expression levels). Thus, the prognostic significance of IL-18 is not mediated by these biological factors [78]. IL-18 may participate in cancer immunity via mechanisms not mediated by TILs or NLRs. Tumor-derived IL-18 has been reported to enhance immunosuppression [79]. In our case, IL-18 showed an inverse increase relative to IL-1β with increasing disease stage, lymphatic vessel involvement, and decreased cancer cell differentiation. Apparently, this is due to the fact that as the disease progresses, this cytokine is redistributed to the lesion site, where its maximum concentration is required.

CCL2 (MCP-1) is involved in the promotion and progression of breast cancer [80]. However, we have shown a decrease in the content of MCP-1 in saliva at advanced stages of breast cancer and with low differentiation of tumors (Figure 3 and Figure 5). It was shown that the content of MCP-1 increases in the presence of expression of HER2, estrogen, progesterone, and low proliferative activity of breast cancer (Figure 6), whereas according to the literature, the expression of MCP-1 in breast carcinomas correlates with the lack of expression of estrogen and progesterone receptors, which indicates a poor prognosis. We appear to observe a negative correlation between serum and salivary MCP-1 in breast cancer. In addition, MCP-1 expression is significantly correlated with the vascular invasion of tumor cells and is closely associated with the expression of matrix metalloproteinase type 1, TNF-α, thymidine phosphorylase, and other angiogenic factors [81]. Overall, current data indicate that increased MCP-1 expression in breast tumors is closely associated with disease progression, but data on MCP-1 levels in body fluids are inconsistent.

We have shown that the content of TNF-α increases in HER2-negative tumors, with negative expression of estrogen receptors and a high index of proliferative activity, i.e., for aggressive subtypes of breast cancer (Figure 6 and Figure 7). An increase in the level of TNF-α was also noted with damage to regional lymph nodes (Figure 4). It is known that coexpression of IL6 and TNF-α was associated with lymph node involvement and shorter survival [82], while RAS and TNF-α together enhanced angiogenesis in breast cancer [83]. Some researchers believe that the effect of TNF-α on breast cancer growth is due not only to its effect on the immune response but also to its ability to induce the expression of angiogenic factors [84]. Activation of TNF-α induces invasive and malignant behavior in breast cancer cells [85], and its overexpression is associated with the regulation of genes and enzymes that mediate estrogen metabolism, leading to higher levels of DNA adducts [86]. Roque showed that the expression levels of inflammatory biomarkers (IL-1β, IL-8, and TNF-α) are associated with DNA damage, which may represent a possible risk for breast cancer [87].

Regarding the expression of VEGF, amino acids (Asp, Cit, and Trp) significantly change their concentration. At the same time, an increase in the content of Asp and Cit is observed. It is known that Cit is a participant in the ornithine cycle in the liver for the synthesis of urea and ornithine and the citrine-NO cycle in the vascular endothelium for the synthesis of nitric oxide (NO) [88]. The substrate for NOS is L-arginine, resulting in the formation of NO and Cit [89,90]. This reaction is cyclic and repeats itself via the interaction of citrulline with aspartate under the influence of arginine succinate synthase to form arginine succinate. Next, arginine succinate lyase catalyzes the formation of arginine succinate into arginine and fumarate. This cycle is anaplerotic since fumarate reacts with water to form malate and is further integrated into the tricarboxylic acid cycle, and L-arginine again acts as a substrate for NOS and the synthesis of NO and Cit [91]. It has been shown that NO under the influence of NOS-2 enhances angiogenesis via the activation of VEGF and suppresses immunity [92].

During the study, we found significant changes in the concentrations of Gln, Gly, and Pro. These amino acids participate in many reactions, which, during metabolic reprogramming of a cancer cell, act as the main source of building materials for the synthesis of proteins and nucleotide bases and also act as antioxidants, as they increase the synthesis of glutathione (GSH) [93,94,95]. Gly [96,97] and Cit [98] have been shown to methylate DNA and histones [99].

## 5. Conclusions

For the first time, we demonstrated changes in the concentration of VEGF and the associated cytokines IL-1β, IL-6, IL-8, and IL-18 in saliva in different molecular biological subtypes of breast cancer depending on the stage of the disease, degree of differentiation, proliferation, and metastasis to lymph nodes. The reasons and nature of multidirectional changes in the concentrations of IL-1β, IL-6, IL-8, and IL-18 are noted and explained from the point of view of redistribution at the systemic and local levels. A correlation has been established between the expression of VEGF and the content of amino acids Asp, Cit, and Trp in saliva. Asp and Cit are involved in the synthesis of the NO signaling molecule, which affects the activity of VEGF. An increase in Trp concentration with a correlation with VEGF reflects an increased need for this amino acid along the IDO pathway to ensure tolerance of the immune system to tumor cells. This combination reflects the fact that VEGF and Trp are vital molecules for the growth of cancer cells. Analysis of the content of VEGF, pro-inflammatory cytokines, and amino acids allows us to understand in more detail the mechanisms of breast cancer progression and subsequently propose more effective therapeutic solutions, taking into account the metabolic characteristics and heterogeneity of breast cancer.

## Figures and Tables

**Figure 1 biomedicines-12-01329-f001:**
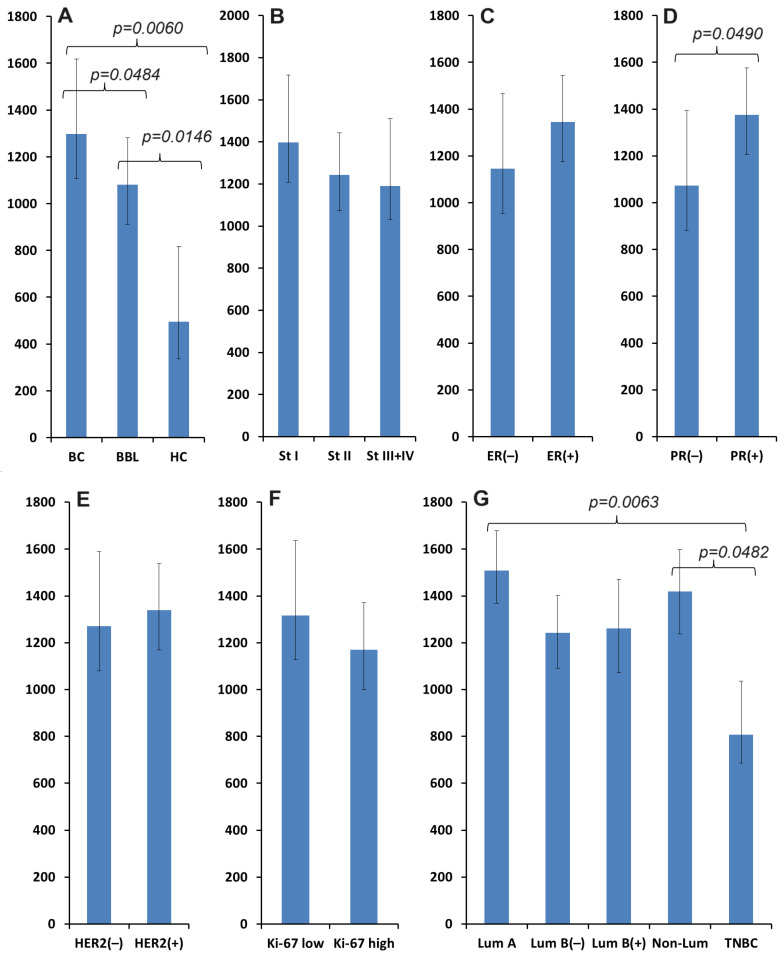
VEGF concentration in saliva (mU/mL): (**A**)—in breast cancer, benign breast pathologies and healthy controls; (**B**)—depending on the stage of breast cancer; (**C**)—depending on the expression of estrogen receptors; (**D**)—depending on the expression of progesterone receptors; (**E**)—depending on the expression of HER2 receptors; (**F**)—depending on the level of the proliferative activity marker Ki-67; (**G**)—depending on the molecular biological subtype of breast cancer.

**Figure 2 biomedicines-12-01329-f002:**
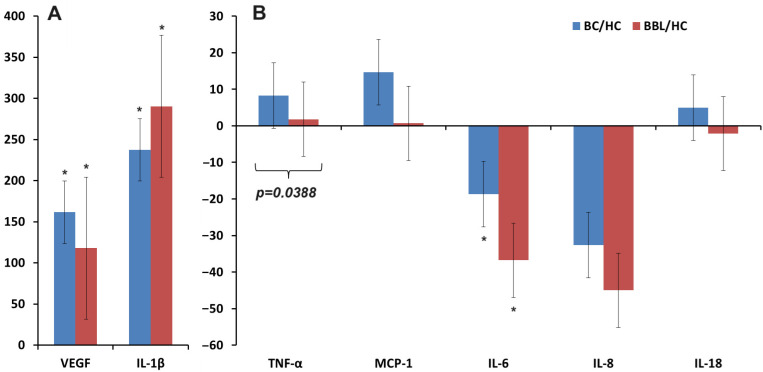
Relative change in the concentration of cytokines in saliva compared to healthy controls (∆, %): (**A**)—VEGF, IL-1β; (**B**)—TNF-α, MCP-1, IL-6, IL-8, IL-18. *—differences with healthy controls are statistically significant (*p* < 0.05); *p*-values are given for statistically significant differences between subgroups. Here and further in Figure 3, Figure 4, Figure 5, Figure 6, Figure 7, Figure 8 and Figure 9, relative changes are calculated as the difference between the corresponding value for breast cancer and healthy controls relative to healthy controls (%). BC—breast cancer, BBL—breast benign lesion, HC—healthy controls.

**Figure 3 biomedicines-12-01329-f003:**
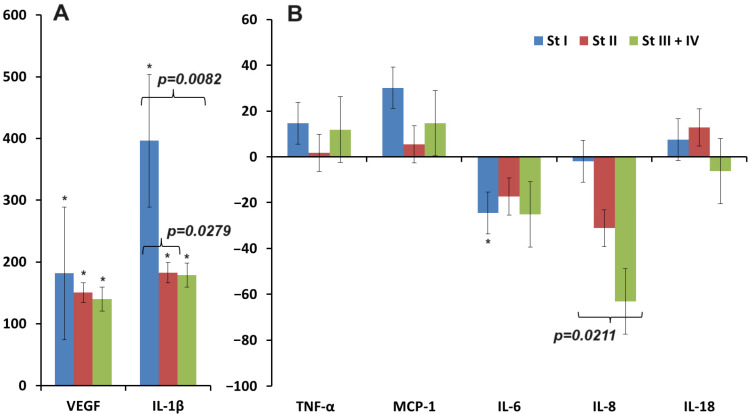
Relative change in the concentration of cytokines in saliva compared to healthy controls depending on the stage of breast cancer (∆, %): (**A**)—VEGF, IL-1β; (**B**)—TNF-α, MCP-1, IL-6, IL-8, and IL-18. *—differences with healthy controls are statistically significant (*p* < 0.05); *p*-values are given for statistically significant differences between subgroups.

**Figure 4 biomedicines-12-01329-f004:**
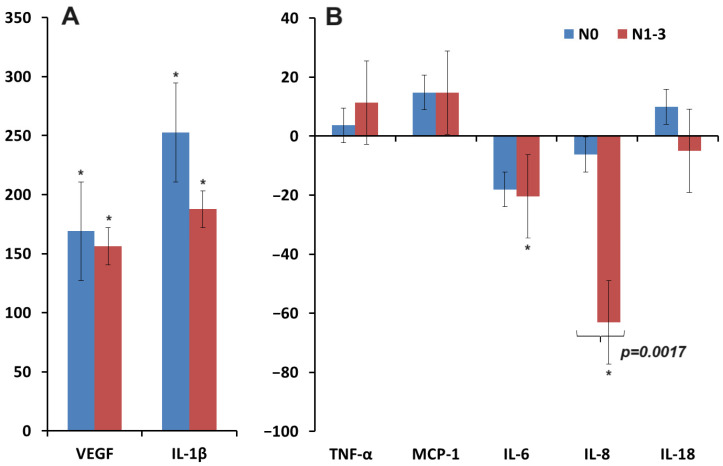
Relative change in the concentration of cytokines in saliva compared to healthy controls depending on the damage to the lymph nodes (∆, %): (**A**)—VEGF, IL-1β; (**B**)—TNF-α, MCP-1, IL-6, IL-8, and IL-18. *—differences with healthy controls are statistically significant (*p* < 0.05); *p*-values are given for statistically significant differences between subgroups.

**Figure 5 biomedicines-12-01329-f005:**
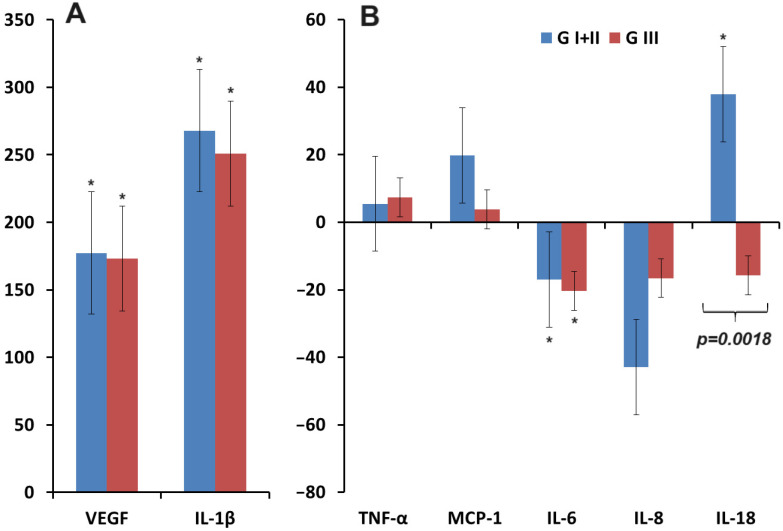
Relative change in the concentration of cytokines in saliva compared to healthy controls depending on the degree of differentiation of breast cancer (∆, %): (**A**)—VEGF, IL-1β; (**B**)—TNF-α, MCP-1, IL-6, IL-8, and IL-18. *—differences with healthy controls are statistically significant (*p* < 0.05); *p*-values are given for statistically significant differences between subgroups. GI—highly differentiated, GII—moderately differentiated, GIII—poorly differentiated breast cancer.

**Figure 6 biomedicines-12-01329-f006:**
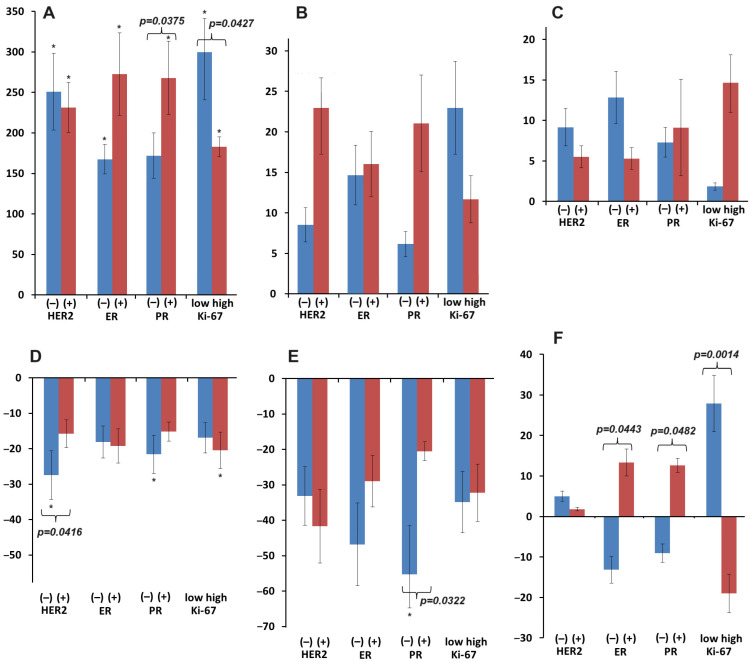
Relative change in salivary cytokine concentrations compared with healthy controls according to HER2, estrogen and progesterone receptor expression status and breast cancer proliferative activity index (∆, %): (**A**)—IL-1β; (**B**)—MCP-1; (**C**)—TNF-α; (**D**)—IL-6; (**E**)—IL-8; (**F**)—IL-18. *—differences with healthy controls are statistically significant (*p* < 0.05); *p*-values are given for statistically significant differences between subgroups.

**Figure 7 biomedicines-12-01329-f007:**
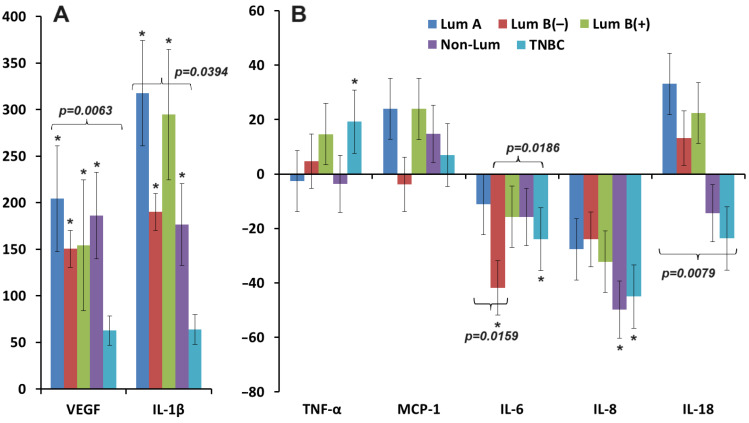
Relative change in the concentration of cytokines in saliva compared to healthy controls depending on the molecular biological subtype of breast cancer (∆, %): (**A**)—VEGF, IL-1β; (**B**)—TNF-α, MCP-1, IL-6, IL-8, and IL-18. *—differences with healthy controls are statistically significant (*p* < 0.05); *p*-values are given for statistically significant differences between subgroups.

**Figure 8 biomedicines-12-01329-f008:**
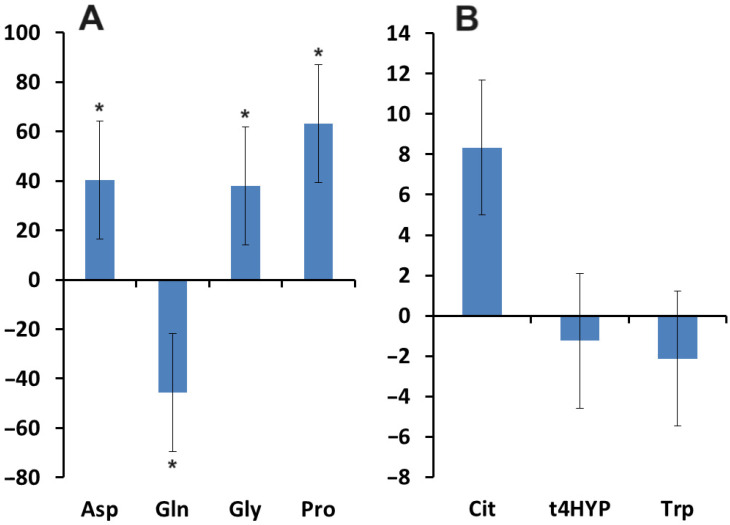
Relative change in salivary amino acid concentrations in breast cancer patients compared with healthy controls (∆, %): (**A**)—Asp, Gln, Gly and Pro; (**B**)—Cit, t4HYP and Trp. *—differences with healthy controls are statistically significant, *p* < 0.05. Asp—aspartic acid, Gln—glutamine, Gly—glycine, Pro—proline, Cit—citrulline, t4HYP—trans-4-hydroxyproline, Trp—tryptophan.

**Figure 9 biomedicines-12-01329-f009:**
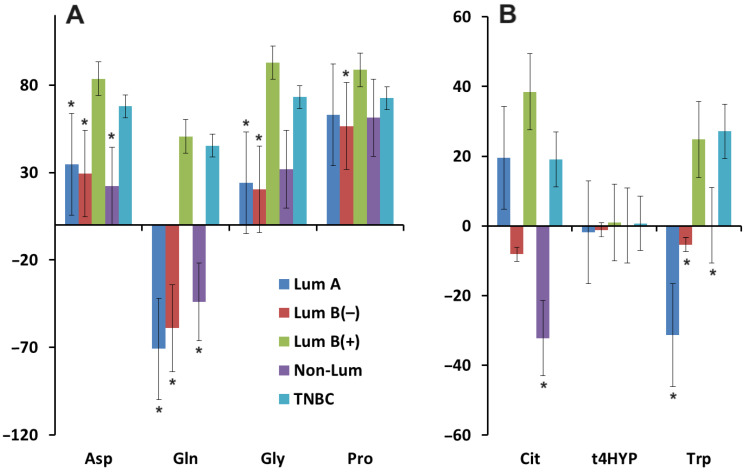
Relative change in salivary amino acid concentrations in breast cancer compared with healthy controls according to molecular biological tumor subtype (∆, %): (**A**)—Asp, Gln, Gly and Pro; (**B**)—Cit, t4HYP and Trp. *—differences with luminal B HER2 (+) breast cancer are statistically significant, *p* < 0.05. Asp—aspartic acid, Gln—glutamine, Gly—glycine, Pro—proline, Cit—citrulline, t4HYP—trans-4-hydroxyproline, Trp—tryptophan.

**Table 1 biomedicines-12-01329-t001:** Characteristics of the study group.

Feature		VEGF and Cytokines	Amino Acids
		n = 230	n = 116
**Clinical Stage**			
	Stage I	76 (33.0%)	37 (31.9%)
	Stage II	95 (41.3%)	43 (37.1%)
	Stage III + IV	59 (25.7%)	36 (31.0%)
**Lymph node status**			
	N_0_	140 (60.9%)	60 (51.7%)
	N_1–3_	90 (39.1%)	56 (48.3%)
**Degree of differentiation (G)**			
	G I + II	104 (45.2%)	74 (63.8%)
	G III	92 (40.0%)	42 (36.2%)
	Unknown	34 (14.8%)	-
**HER2 status**			
	HER2-negative	159 (69.1%)	88 (75.9%)
	HER2-positive	62 (27.0%)	28 (24.1%)
	Unknown	9 (3.9%)	-
**Estrogen (ER) status**			
	ER-negative	73 (31.7%)	26 (22.4%)
	ER-positive	149 (64.8%)	90 (77.6%)
	Unknown	8 (3.5%)	-
**Progesterone (PR) status**			
	PR-negative	103 (44.8%)	46 (39.7%)
	PR-positive	119 (51.7%)	70 (60.3%)
	Unknown	8 (3.5%)	-
**Ki-67**			
	low (<20%)	117 (50.9%)	59 (50.9%)
	high (>20%)	93 (40.4%)	57 (49.1%)
	Unknown	20 (8.7%)	-
**Subtype**			
	Luminal A-like	61 (26.5%)	40 (34.5%)
	Luminal B-like (HER2−)	57 (24.9%)	35 (30.2%)
	Luminal B-like (HER2+)	33 (14.3%)	15 (12.9%)
	HER2-enriched (Non-Lum)	30 (13.0%)	12 (10.3%)
	Triple-negative	41 (17.8%)	14 (12.1%)
	Unknown	8 (3.5%)	-

**Table 2 biomedicines-12-01329-t002:** Average content of cytokines in the saliva of the study groups.

Cytokines	Breast Cancer, n = 230	Breast Benign Lesion, n = 92	Healthy Controls, n = 59	Kruskal–Wallis Test; *p*-Value
**VEGF, mU/mL**	1297.4 [586.7; 2119.5]	1081.1 [505.4; 1667.4]	496.1 [352.7; 1360.2]	10.20; 0.0061 *
**IL-1β, pg/mL**	124.8 [30.84; 305.4]	144.5 [37.00; 310.1]	37.01 [11.78; 106.1]	8.796; 0.0123 *
**TNF-α, pg/mL**	2.63 [1.98; 3.70]	2.47 [1.33; 3.02]	2.43 [2.07; 3.32]	4.465; 0.1073
**MCP-1, pg/mL**	52.93 [35.43; 100.1]	46.46 [23.75; 88.43]	46.14 [35.43; 120.1]	2.573; 0.2762
**IL-6, pg/mL**	3.57 [2.22; 5.08]	2.78 [1.54; 5.96]	4.39 [2.78; 6.63]	4.181; 0.1236
**IL-8, pg/mL**	68.30 [24.83; 141.3]	55.73 [21.22; 183.1]	101.3 [22.73; 187.1]	0.5910; 0.7442
**IL-18, pg/mL**	67.05 [31.13; 132.9]	62.50 [40.90; 134.3]	63.86 [22.50; 141.8]	0.3925; 0.8218

Note. *—differences between the three groups are statistically significant, *p* < 0.05.

## Data Availability

All data and materials used in this study are available from the corresponding author due to (the data is part of the dissertation research) and will be provided upon reasonable request.

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
