# Peer review of "The Role of Salivary Vascular Endothelial Growth Factor A, Cytokines, and Amino Acids in Immunomodulation and Angiogenesis in Breast Cancer"

_biomedicines, 2024, doi:10.3390/biomedicines12061329_

Round 1

Reviewer 1 Report

Comments and Suggestions for Authors

Sarf et al. analyzed the salivary VEGF, cytokines, and amino acids content in different breast cancer patients and healthy control groups and explored the correlation between them. The authors first compared the salivary VEGF, cytokines, and amino acids between breast cancer patients and healthy controls in different breast cancer subtypes. Then the authors analyzed the correlation between VEGF, cytokines, and amino acids in different breast cancer subtypes. This is a good study report on salivary VEGF, cytokines, and amino acid contents. However, the manuscript needs extensive text and figure editing. The introduction of the cytokines studied in their work is not enough. Maybe, some parts of the discussion section can be moved to the introduction section. Nevertheless, the following issues should be resolved to improve the presentation of the manuscript.

Comments:

1.                         Lines 58-60: This sentence is not quite logical. Please rephrase.

2.                         Lines 72-75: This part is quite confusing. Do the 230 patients include healthy controls? If yes, please clarify this in the main text. If not, and 230 is only the number of breast cancer patients, please include information about healthy controls and patients with benign breast disease. Table 1 only shows the information of 230 breast cancer patients with VEGF and cytokines measurements.

3.                         Section 3.1, lines 139-155: please mention each panel of Figure 1 separately in the main text. The current description is too wordy without even mentioning which figure panel each statement refers to.

4.                         All figures:

a.      Please keep the same cases of all figure panel labels in the figure, figure legend, and main text the same.

b.     Please keep the location of the figure panel labels consistent and make sure they are clearly seen and easier to know what they refer to.

c.      Please make sure the p-value labels are easier to know what they refer to. Current p-value labels are hard to read and look unprofessional.

5.                         Figure 2 and Figure 9: please define the relative changes. How do you calculate them? I guess relative change equals (breast cancer-cancer-healthy control) / healthy control?

6.                         Line 198: Please explain GI+II and GIII here and in the Figure 5 legend.

7. Sections 3.1 to 3.2: The paragraphs in these two sections need a brief introduction and transition words to make the main text smoother.

8.                         Section 3.2 and 3.3: Please add bar graphs to show the overall average component of the salivary cytokines and amino acids, respectively. What are the percentages of each cytokine or amino acid in these groups?

9.                         Section 2, lines 163-167 and section 3, lines 234-239: Please include some kinds of figures or illustrations with raw data points to show these calculated correlation coefficients.

Reviewer 2 Report

Comments and Suggestions for Authors

This article describes the levels of VEGF, cytokines and amino acids in the saliva of breast cancer patients. The results are well presented and conclusions are appropriate.

Concern:

1. The authors should add more detailed comparisons regarding levels of VEGF, cytokines and amino acids in blood, tumor and saliva, preferably adding a table for the readers.

2. The authors should test levels of different isoforms of VEGF.

Round 2

Reviewer 1 Report

Comments and Suggestions for Authors

The authors revised the manuscripts according to reviewers’ comments. I appreciate that the authors treat the reviewers’ comments seriously. From the scientific point of view, I have no more concerns although I still think the figures need more work. For example, the panel labels in each figure should be put at the top left corner of each figure panel. Nevertheless, this is a good manuscript and can be accepted after resolving the small figure issues.
